# Microbiological Monitoring and Microbial Susceptibility of *Salmonella* from Aquacultured Tambaqui Hybrids (*Colossoma macropomum*): Implications for Food Safety

**DOI:** 10.3390/antibiotics14101047

**Published:** 2025-10-19

**Authors:** Cristiane Coimbra de Paula, Yuri Duarte Porto, Fabiola Helena dos Santos Fogaça, Wagner de Souza Tassinari, Vinícius Silva Castro, Adelino Cunha-Neto, Ricardo César Tavares Carvalho, Luciana Kimie Savay-da-Silva, Eduardo Eustáquio de Souza Figueiredo, Tathiana Ferguson Motheo

**Affiliations:** 1Postgraduate Program in Animal Bioscience, University of Cuiabá (UNIC), Cuiabá 78065-900, Mato Grosso, Brazil; cristiane.paula@univag.ed.br (C.C.d.P.); adelino.neto@ufmt.br (A.C.-N.); ricardo.ct.carvalho@cogna.com.br (R.C.T.C.); tathiana.motheo@cogna.com.br (T.F.M.); 2Department of Food Science and Technology, Federal University of Mato Grosso (UFMT), Cuiabá 78060-900, Mato Grosso, Brazil; ydporto@ufrrj.br (Y.D.P.); v.castro@uleth.ca (V.S.C.); luciana.silva@ufmt.br (L.K.S.-d.-S.); 3Embrapa Food Technology, Rio de Janeiro 23020-470, Rio de Janeiro, Brazil; fabiola.fogaca@embrapa.br; 4Clinical Epidemiology Laboratory, Evandro Chagas National Institute of Infectious Diseases, Oswaldo Cruz Foundation (FIOCRUZ), Rio de Janeiro 21040-361, Rio de Janeiro, Brazil; wagner.tassinari@ini.fiocruz.br; 5Department of Mathematics, Institute of Exact Sciences, Federal Rural University of Rio de Janeiro, Seropédica 23890-000, Rio de Janeiro, Brazil

**Keywords:** aquatic food safety, fish farming, fish-borne pathogens, non-typhoidal *Salmonella*, sanitary surveillance, zoonosis

## Abstract

**Background:** Salmonellosis is a foodborne illness typically associated with gastroenteritis following the ingestion of products contaminated with *Salmonella enterica*. Although the aquatic environment is not a natural reservoir for *Salmonella* spp., its occurrence has been reported in various aquacultured species worldwide, including species from the Amazon Basin in South America. The World Health Organization has classified the emergence of multidrug-resistant (MDR) *Salmonella* strains as a global priority, underscoring the importance of monitoring antimicrobial resistance to mitigate public health risks. This study aimed to detect *Salmonella* spp. serotypes of clinical relevance to humans (*S.* Typhi, *S.* Paratyphi, *S.* Typhimurium, and *S.* Enteritidis) in farmed tambaqui hybrids and to assess the antimicrobial susceptibility of the isolates. **Methods:** A total of 55 *Salmonella* spp. strains, previously isolated from tambaqui hybrids (*Colossoma macropomum*) produced in fish farms in Mato Grosso, Brazil, were evaluated. Identification and susceptibility profiling were performed using the VITEK^®^2 Compact automated system (BioMérieux, Marcy l’Étoile, France), testing 14 commonly used antimicrobials, including amoxicillin–clavulanic acid, piperacillin–tazobactam, cephalexin, cefuroxime axetil, ceftriaxone, cefepime, meropenem, ertapenem, amikacin, gentamicin, ciprofloxacin, and sulfamethoxazole–trimethoprim. **Results:** All isolates were confirmed as *Salmonella* spp., with no detection of clinically important serotypes. Moreover, all 55 strains were susceptible to the 14 antimicrobials tested. **Conclusions:** These findings indicate a low risk of pathogenic or resistant *Salmonella* from farmed tambaqui hybrids under the evaluated conditions. Nevertheless, ongoing microbiological monitoring remains essential, particularly in light of regulatory standards that prohibit the presence of *Salmonella* spp. in fish products and the potential emergence of MDR strains.

## 1. Introduction

*Salmonella* is one of the most common foodborne zoonotic pathogens and poses a significant threat to global public health. Its prevalence is particularly concerning given that *Salmonella* is considered the most frequent foodborne organism in products imported from Africa to the European Union [1]. In the United States, the two most common serovars are *S*. Typhimurium and *S*. Enteritidis, which together account for 41.5% of all reported outbreaks [2]. Furthermore, these two serovars represent almost 60% of all *Salmonella* outbreaks worldwide [3] and are responsible for 91% of cases in Africa [4], underscoring their widespread dissemination and impact on global public health.

Fish consumption in Brazil remains below the global average, with a per capita intake of approximately 10 kg per year, compared to the worldwide average of 20.5 kg per year. However, national aquaculture output has shown consistent expansion, evidenced by a 3.1% increase in production in 2023 [5,6]. Within this context, the state of Mato Grosso emerges as a leading producer, benefiting from its extensive water resources and strategic location across the Amazon, Platina, and Tocantins river basins [7,8]. The state hosts a rich diversity of native fish species of nutritional, economic, and zootechnical importance [5,8]. Notable among these are the pintado (*Pseudoplatystoma corruscans*), valued in sport fishing and regional cuisine; the piracanjuba (*Brycon orbignyanus*), recognized for its meat quality; and the tambatinga (*Colossoma macropomum* × *Piaractus brachypomus*), an extensively cultivated hybrid in the Amazon and Central-West regions, recognized as a key species in Brazilian aquaculture. Other relevant species include the tambaqui (*Colossoma macropomum*) and the dourado (*Salminus brasiliensis*)*,* all well adapted to the region’s hydroclimatic conditions [9,10].

Beyond fish and seafood, Mato Grosso is a major producer of other meat matrices, such as beef (Brazil’s largest producer), poultry (the 7th largest nationally), and pork (the 6th largest nationally) [11]. In addition, aquaculture is frequently co-located with other meat production, strengthening concerns about *Salmonella* cross-contamination [12].

Accordingly, the intensification of fish farming practices and the co-location of fish farming with other animal production systems have raised concerns about food safety, particularly regarding the presence of pathogenic bacteria. Studies conducted in Mato Grosso have reported the occurrence of *Salmonella* spp. in native farmed fish, especially tambatinga. Several serotypes of public health significance have been identified, including *S.* Heidelberg, *S.* Saintpaul, *S.* Panama, *S.* Swarzengrund, *S.* Abony, *S.* Ndolo, *S.* Mbandaka, *S.* Rough, *S.* O:16, and *S.* Typhimurium [13,14,15,16]. These findings underscore the importance of continuous microbiological surveillance in aquaculture systems.

The emergence of multidrug-resistant strains of *Salmonella* spp. represents an additional public health challenge. Classified as a priority by the World Health Organization (WHO), these strains are associated with therapeutic failure, prolonged hospitalizations, and increased mortality [17,18,19].

Antimicrobial resistance (AMR) is a pressing global public health challenge that necessitates coordinated action across multiple sectors [20]. Recognizing this, the Pan American Health Organization (PAHO) actively supports the fight against AMR through capacity building, fostering international cooperation, and advancing research efforts. These multisectoral initiatives, grounded in scientific evidence, are crucial for preventing and responding to AMR threats, while also promoting awareness and developing effective policies [1,10].

Building on the importance of a comprehensive approach to public health, the promotion of food safety is equally vital in mitigating health risks such as foodborne diseases, including human salmonellosis [10]. Effective control of these diseases relies on the awareness and rigorous implementation of food safety pillars. Engaging all stakeholders’ consumers, farmers, markets, and others involved in the food system, through formal and informal education about basic food safety practices is essential. Together, these efforts in antimicrobial resistance management and food safety promotion form a cohesive strategy to protect and improve public health at the global level [1,10,20].

In intensive aquaculture systems, the recurrent use of antimicrobials, such as oxytetracycline, florfenicol, sulfonamides, enrofloxacin, and tetracycline, particularly when applied for prophylactic purposes, has been identified as a contributing factor to the selection of resistant bacterial strains, reducing treatment effectiveness and increasing risks to consumer safety [21,22,23]. Among the available tools for microbial identification and antimicrobial susceptibility testing, automated systems such as the Vitek^®^2 Compact are widely used in both clinical and food microbiology [24] and have also been successfully applied in studies involving foodborne *Salmonella* [25]. This supports the relevance of employing this approach in the present study.

Considering the potential public health risks, this study aimed to identify clinically relevant *Salmonella* spp. strains isolated from tambatinga (*Colossoma macropomum* × *Piaractus brachypomus*) farmed in Mato Grosso, Brazil, a region spanning the Cerrado and Pantanal biomes, and to assess their antimicrobial susceptibility profiles to antibiotics commonly used in the treatment of salmonellosis.

## 2. Results

Based on the biochemical identification provided by the VITEK^®^2 system, none of the isolates were presumptively identified as clinically relevant serotypes such as *Salmonella* Typhi, *S.* Paratyphi, *S.* Typhimurium, or *S.* Enteritidis. However, since no confirmatory serological typing was performed, all isolates remained classified at the genus level (*Salmonella* spp.). Importantly, the genus-level identification provided by VITEK^®^2 was consistent with the previous confirmation of the isolates by PCR targeting the *hilA* gene, as described in the Section 4. Antimicrobial susceptibility testing revealed that all isolates were susceptible to the antimicrobial classes evaluated, including beta-lactams, carbapenems, aminoglycosides, fluoroquinolones, phosphonic acid derivatives, and sulfonamides. All 55 *Salmonella* spp. strains exhibited an identical susceptibility profile (Table 1).

The values presented in Table 1 correspond to the minimum inhibitory concentration (MIC) obtained for each isolate using the VITEK^®^2 Compact system. As the MIC represents the lowest concentration of an antimicrobial agent that prevents visible growth of the microorganism, it is reported together with the categorical interpretation (S, I, R) according to BrCAST guidelines [26]. Therefore, these values are not average across replicates and do not yield measures of variability such as standard error or standard deviation.

The MIC results showed a consistent pattern across all 55 strains, with full susceptibility to the antimicrobials tested: amoxicillin/clavulanic acid; piperacillin/tazobactam; cephalexin, cefuroxime axetil, ceftriaxone, cefepime, fosfomycin, meropenem, ertapenem, amikacin, gentamicin, ciprofloxacin, trimethoprim/sulfamethoxazole. 

Only cefuroxime was interpreted as “susceptible, increased exposure.” This behavior may be explained by its pharmacological characteristics: cefuroxime axetil is an orally administered ester prodrug that is hydrolyzed in vivo to the active form cefuroxime, while the sodium form is typically administered intravenously.

## 3. Discussion

The absence of *Salmonella* Typhi, *S.* Paratyphi, *S.* Typhimurium, and *S.* Enteritidis in the analyzed samples is a positive indicator from a food safety perspective, as these serotypes are frequently associated with severe cases of human salmonellosis [19,21]. It is important to note that the samples originated from non-commercialized fish, which may have contributed to the absence of these clinically relevant serotypes. However, previous reports have demonstrated that such serotypes can be present in products available in markets and consumed by the general population [27,28]. Despite the non-detection of these pathogenic strains, the presence of *Salmonella* spp. highlights the need to implement rigorous sanitary controls, particularly in the early stages of the production chain, including cultivation environments where contamination can easily occur.

Previous studies support the variability in *Salmonella* serotype distribution over time, across regions, and along the aquaculture chain worldwide [9,10,13]. In Brazil, ref. [14] identified serotypes such as *S*. Typhimurium, *S.* Ndolo, *S*. Mbandaka, *S*. Rough, and *S*. O:16 in a slaughterhouse in Mato Grosso, while ref. [29] also detected *S*. Typhimurium in Rio de Janeiro. Although serotypes like *S.* Typhi, *S*. Paratyphi, *S.* Typhimurium, and *S*. Enteritidis are commonly associated with systemic infections, other serotypes have been linked to gastroenteritis, as reported by [30,31]. In Mato Grosso State, variation in serotypes has also been described in studies addressing both production and commercialization stages. *S*. Heidelberg, *S*. Saintpaul, and *S.* Panama were identified in fish farming environments [16], while *S.* Swarzengrund and *S*. Abony were isolated from commercialized products [15]. Consistent with these findings, the present study, which focused on fish from breeding tanks, did not detect the classic pathogenic serotypes.

The occurrence of *Salmonella* spp. in fish farming and processing environments reflects a potential contamination risk, often associated with poor handling practices [32]. This contamination may result from exposure to fecal matter from wild animals, human sewage, or other environmental pollutants [33,34]. Although fish are not considered natural reservoirs of *Salmonella*, the detection of multiple serotypes suggests a significant environmental influence on the microbiological quality of aquaculture products [13]. In a study conducted by Deaven et al. [35], the authors demonstrated the high genomic diversity of *Salmonella* in freshwater systems, where seasonality appears to exert a stronger influence, through both rainfall events and patterns of animal reservoirs. The authors reinforced that once the water is contaminated the bacterium may adopt stress-mitigating strategies such as biofilm formation, association with algae, or entering a viable but non-culturable (VBNC) state.

Also, the need to adapt to environmental stress has been associated with attenuation of virulence and antimicrobial-resistance responses, described as a trade-off effect in recent reports [36]. These environmental pressures may underlie the consistently low antibiotic-resistance findings in aquaculture, which contrast with those observed in other meat matrices. In line with this, ref. [37] suggested that adaptability to aquatic environments may favor survival strategies that do not rely on antibiotic resistance, such as biofilm formation or horizontal gene transfer, depending on the selective pressures present. The authors further emphasized that stricter regulations and enforcement can reduce selective pressure, favoring the persistence of susceptible strains.

The isolates analyzed in the present study demonstrated full susceptibility to all antimicrobials tested, representing a favorable outcome. However, the lack of observed resistance does not eliminate the potential for future risk associated with consumption. The ‘I’ interpretation for cefuroxime may be linked to pharmacological factors; however, this finding is in stark contrast to studies like that of [25], which documented resistance to the same antimicrobial. The results vary considerably from those reported in that study, even though the same susceptibility profiling technology (Vitek^®^2) was employed. While our isolates were susceptible to all antimicrobials tested, ref. [25] found multidrug resistance in all 27 *Salmonella* strains isolated from mussels, including resistance to cefuroxime and cefuroxime-axetil, both second-generation cephalosporins. In addition, they identified only *Salmonella* spp., similar to our findings.

Other studies using the Vitek^®^2 system have also reported divergent resistance patterns. Ref. [38] analyzed 657 animal feed samples and isolated 80 *Salmonella* spp. strains, reporting resistance to cephalexin (a first-generation cephalosporin) and increased susceptibility to cefovecin (a third-generation cephalosporin), although cefuroxime was not tested. Similarly, ref. [39] detected *Salmonella* in 13 of 125 samples analyzed, identifying *S.* Typhimuraium (cattle), *S.* Montevideo (poultry), and *S.* Kentucky (poultry), in addition to some non-typeable strains from soil, cattle, and poultry. In contrast to the present study, ref. [39] reported resistance to cefoxitin (a second-generation cephalosporin) across all isolates, further illustrating that *Salmonella* populations from different matrices and production systems exhibit highly variable susceptibility profiles.

This variation emphasizes the significance of localized surveillance and indicates that, in the study region, resistance mechanisms are not yet fully developed, in contrast to other aquatic environments. The misuse or overuse of antimicrobial agents in aquaculture may promote the selection and spread of resistant strains [40,41]. Intensified fish production practices have led to an increased prevalence of illnesses. The application of veterinary drugs in aquaculture has emerged as a strategy to ensure animal health and facilitate large-scale production [42]. The antibiotics approved for aquaculture use by the relevant authorities include oxytetracycline, florfenicol, and the sulfadimethoxine–ormetoprim combination, though specific uses are highly restricted [43]. In Brazil, aquaculture currently is restricted to two licensed active ingredients: florfenicol and oxytetracycline [44]. Nonetheless, florfenicol is the predominant agent employed in the management of bacterioses. The susceptibility tests for florfenicol and oxytetracycline were not performed. Our purpose was to assess the susceptibility of aquaculture strains to antimicrobials from classes deemed critically important for treating human salmonellosis. From this viewpoint, complete susceptibility to these categories is a highly favorable indicator for public health. Importantly, Mohammed et al. [45] highlighted that the use of probiotics, self-medication practices, and the availability of antibiotics in fish farming vary regionally. In areas with stricter regulations, lower prevalence of resistance has been reported, whereas more permissive contexts may show higher resistance rates. These differences reinforce the impact of regulatory frameworks and local management practices on observed resistance profiles.

This concern aligns with the One Health approach, which recognizes the interconnectedness of human, animal, and environmental health, underscoring the importance of continuous surveillance and responsible antimicrobial use [46,47,48,49].

Although antimicrobial resistance is a recognized global health threat, the findings of this study revealed a contrasting scenario. All antimicrobial classes tested, including trimethoprim–sulfamethoxazole, ciprofloxacin, and ceftriaxone, were effective against the *Salmonella* spp. strains. These results are particularly relevant given the frequent use of these antimicrobials in the treatment of salmonellosis. Nonetheless, it is essential to consider that the use of antibiotics in human and veterinary medicine, as well as in aquaculture, may lead to residual contamination in animal-derived foods. Such residues contribute to the selection of resistant commensal bacteria and the spread of resistance genes [29,31].

An important point is that multidrug-resistant *Salmonella* has already been documented in the state of Mato Grosso in poultry [50] and in beef [51], in contrast to the present study. However, in a study conducted by [14], the authors investigated contamination along the slaughter line of a fish-processing plant, and the isolates were susceptible to the antibiotics tested. In the present study, we expanded both the sample size and the number of fish farms assessed and confirmed the susceptibility profile of the isolates.

Resistance genes to various classes of antimicrobials have been documented in *Salmonella* spp. [52,53,54]. The transmission of these genes may occur through direct contact with animals or indirectly via food, water, or agricultural waste [55,56,57]. Although the present study did not detect strains of clinical concern, previous research by [58] reported *Salmonella* Weltevreden resistant to sulfamethoxazole–trimethoprim. In contrast, the isolates from tambatinga in this study showed full susceptibility, reinforcing the continued efficacy of this antimicrobial combination in treating *Salmonella* infections.

Given its critical role in managing severe bacterial infections, ceftriaxone, a third-generation cephalosporin, requires vigilant surveillance, particularly because resistance has been associated with adverse clinical outcomes, including increased morbidity and mortality. The susceptibility of all tested strains to this agent is a relevant finding with implications for both clinical treatment and public health policy. Nevertheless, previously reported resistance patterns highlight the ongoing need for active and systematic monitoring of antimicrobial susceptibility.

Differences in serotype distribution and resistance profiles across regions and production systems have been well documented, emphasizing the importance of continuous local surveillance [59,60]. Even in the absence of classical pathogenic serotypes, the presence of *Salmonella* spp. in aquaculture products represents a persistent public health risk, particularly due to the challenges in early detection, especially in non-commercialized products. Per the current Brazilian regulations, *Salmonella* spp. is not permitted in ready-to-eat foods, as established by Normative Instruction No. 161 of 1 July 2022 [61], reinforcing the need for stringent control measures at all stages of the production chain.

## 4. Materials and Methods

### 4.1. Salmonella spp. Isolates

A total of 55 *Salmonella* spp. strains isolated from tambatinga (*Colossoma macropomum* × *Piaractus brachypomus*) were analyzed. These strains were originally obtained and characterized as part of a doctoral research project [62], from samples of 72 tambatinga fish (average weight 1.3 kg) analyzed as pooled samples, in which tissues such as scales, gills, and viscera (esophagus, stomach, liver, intestine, and feces) were examined. Isolation followed the ISO 6579-1 protocol [63], including pre-enrichment in Buffered Peptone Water (BPW; Kasvi, Curitiba, Brazil), selective enrichment in Rappaport-Vassiliadis (RVS; Kasvi, Curitiba, Brazil) and Tetrathionate (TT; HiMedia Brasil, Campinas, Brazil) broths, and plating on Xylose Lysine Deoxycholate (XLD; Neogen do Brasil, Indaiatuba, Brazil) and Brilliant Green Agar (BGA; BD Difco, São Paulo, Brazil). Up to five typical colonies were purified on Nutrient Agar (Kasvi, Curitiba, Brazil) and screened by biochemical tests using Triple Sugar Iron Agar (TSI; HiMedia Brasil, Campinas, Brazil) and Lysine Iron Agar (LIA; Kasvi, Curitiba, Brazil). Colonies with presumptive positive results were subjected to genus-level confirmation by PCR targeting the *hilA* gene (497 bp) [64]. PCR products were visualized on 1.5% agarose gel, and no sequencing was performed. Confirmed isolates were cryopreserved at −80 °C until further analysis.

For the present study, strains were reactivated from cryopreservation in Brain Heart Infusion (BHI; Kasvi, Curitiba, Brazil) broth and subsequently maintained on Nutrient Agar (Kasvi, Curitiba, Brazil) prior to antimicrobial susceptibility testing. They originated from 25 fish farms located in eight municipalities of the Baixada Cuiabana region, Mato Grosso State, Brazil. These included Chapada dos Guimarães, Cuiabá, Jangada, Rosário Oeste, and Várzea Grande, situated in the Cerrado biome (similar to a savanna biome), and Nossa Senhora do Livramento, Poconé, and Santo Antônio do Leverger, situated in the Pantanal biome (similar to a wetland biome) [65]. Sampling was performed during both the rainy season (November to April) and the dry season (May to October) [66] between 2022 and 2023. Further methodological details are available in the original doctoral study [62].

### 4.2. Identification of Serotypes of Human Clinical Importance

To identify potentially clinically relevant serotypes, 55 *Salmonella* spp. strains were reactivated from cryopreservation in BHI broth (Kasvi, Curitiba, Brazil) and subsequently maintained on Nutrient Agar (Kasvi, Curitiba, Brazil). Following microbial growth, biochemical identification was performed using carbon source utilization and enzymatic activity profiles via the Vitek^®^ 2 Compact System (BioMérieux, Marcy l’Étoile, France) [24]. The GN identification card (Gram-Negative Bacteria REF 21341; BioMérieux 044066-05; 2021-05) was employed to differentiate glucose-fermenting from non-fermenting bacteria. According to the manufacturer, the system can identify species such as *Salmonella enterica* subsp. enterica and certain serovars (*S*. Enteritidis, *S*. Paratyphi B, *S*. Paratyphi C, and *S*. Typhimurium); however, serotype confirmation requires complementary serological testing [67].

### 4.3. Antimicrobial Susceptibility Testing

Antimicrobial susceptibility testing of *Salmonella* spp. isolates was conducted using the VITEK^®^ 2 System (BioMérieux, Marcy l’Étoile, France), which applies a semiquantitative method to determine susceptibility and resistance profiles, as well as extended-spectrum β-lactamase (ESBL) phenotypes, as described by [68]. The AST-408 panel (REF 423924; BioMérieux, Marcy l’Étoile, France) was employed, containing 14 antimicrobials from seven pharmacological classes. These included the penicillin’s amoxicillin–clavulanic acid (AMC) and piperacillin-tazobactam (PPT); the beta-lactams cephalexin (CFE), cefuroxime (CRX), cefuroxime axetil (CRXA), ceftriaxone (CRO), and cefepime (CPM); the carbapenems ertapenem (ERT) and meropenem (MPM); the aminoglycosides amikacin (AMI) and gentamicin (GENT); the fluoroquinolone ciprofloxacin (CIP); the phosphonic acid derivative fosfomycin (FOS); and the sulfonamide trimethoprim–sulfamethoxazole (SUT).

The concentrations of each antimicrobial followed the manufacturer’s standards. Results were interpreted according to the Brazilian Committee on Antimicrobial Susceptibility Testing (BrCAST) guidelines [26] for the *Enterobacteriaceae* group, which are harmonized with the European Committee on Antimicrobial Susceptibility Testing (EUCAST). These guidelines establish the categorical thresholds for interpretation: susceptible (S), susceptible with increased exposure (I), or resistant (R). The specific breakpoints applied for each antimicrobial in this study were as follows: AMC (S ≤ 8 µg/mL, R ≥ 16 µg/mL); PPT (S ≤ 8, R ≥ 16); CFE (S ≤ 16, R ≥ 32); CRX (S ≤ 0.001, I = 0.002–8, R ≥ 8); CRXA (S ≤ 8, R ≥ 8); CRO (S ≤ 1, I = 2, R ≥ 2); CPM (S ≤ 1, I = 2–4, R ≥ 4); FOS (S ≤ 8, R ≥ 8); SUT (S ≤ 2, I = 4, R ≥ 4); MPM (S ≤ 2, R ≥ 8); ERT (S ≤ 0.5, R ≥ 0.5); AMI (S ≤ 8, R ≥ 8); GENT (S ≤ 2, R ≥ 2); CIP (S ≤ 0.06, R ≥ 0.06). Isolates exhibiting resistance to three or more antimicrobial classes were classified as multidrug-resistant (MDR) [23,69].

## 5. Conclusions

In this study, no *Salmonella* spp. serotypes associated with clinically significant foodborne illness were identified. Although the consumption of native regional fish may present a potential risk for salmonellosis, the findings suggest a low likelihood of public health impact, as all isolated strains were susceptible to the tested antimicrobials, indicating a probable absence of prior exposure to antibiotic agents. Despite this favorable outcome, current legislation mandates the absence of *Salmonella* spp. in fish products to ensure both animal health and environmental safety.

The detection of *Salmonella* spp. underscores the importance of epidemiological surveillance and traceability throughout the aquaculture production chain. These measures are critical for enabling timely and effective responses in the event of contamination or outbreaks. Accordingly, future studies adopting a One Health perspective are essential for improving the understanding of foodborne pathogens, monitoring emerging resistance patterns, and informing strategies to safeguard public health.

## Figures and Tables

**Table 1 antibiotics-14-01047-t001:** Antimicrobial susceptibility profiles of 55 *Salmonella* spp. strains isolated from tambatinga (*Colossoma macropomum* × *Piaractus brachypomus*) from eight municipalities in the Baixada Cuiabana region, Mato Grosso, Brazil.

Antibiotic	MIC	Analysis	Antibiotic	MIC	Analysis
**Penicillins**			**Sulfa**		
Amoxicillin/ Clavulanic acid	≤2	S	Trimethoprim/ Sulfamethoxazole	≤20	S
Piperacillin/ Tazobactam	≤4	S	**Carbapenems**		
**Beta-lactams**			Meropenem	≤0.25	S
Cephalexin	≤4	S	Ertapenem	≤0.12	S
Cefuroxime	4	I	**Aminoglycosides**		
Cefuroxime Axetil	4	S	Amikacin	≤1	S
Ceftriaxone	≤0.25	S	Gentamicin	≤1	S
Cefepime	≤0.12	S	**Fluoroquinolones**		
**Phosphonics**			Ciprofloxacin	≤0.06	S
Fosfomycin	≤0.16	S			

Minimum inhibitory concentration (MIC, expressed in µg/mL) values were determined using the VITEK^®^2 Compact system (BioMérieux, France), and results were interpreted according to the Brazilian Committee on Antimicrobial Susceptibility Testing (BrCAST) [26]. Legend: S = susceptible; I = susceptible, increased exposure.

## Data Availability

The raw microbiological data and detailed isolation procedures for the *Salmonella* spp. strains analyzed in this study are available in the doctoral dissertation of Porto (2023), publicly accessible at the institutional repository of the Federal Rural University of Rio de Janeiro: https://tede.ufrrj.br/jspui/handle/jspui/6673 (accessed on 29 October 2023). Additional datasets generated and analyzed during the current study (antimicrobial susceptibility results) are available from the corresponding author upon reasonable request.

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
