# Peer review of "Microbiological Monitoring and Microbial Susceptibility of Salmonella from Aquacultured Tambaqui Hybrids (Colossoma macropomum): Implications for Food Safety"

_antibiotics, 2025, doi:10.3390/antibiotics14101047_

Round 1

Reviewer 1 Report

Comments and Suggestions for Authors

In the present study, the authors investigated the food safety issue by monitoring microbial susceptibility of Salmonella in tambaqui hybrid. Since the present manuscript is supported by only one Table, major concerns need to be addressed or clarified before being acceptable for publication. Below, you can find some comments:

  1. Please specify the Table 1 data, antimicrobial susceptibility profile of 55 Salmonella strains should have a standard error if these MIC values are presented as average number.
  2. Section 4, please provide more detailed information for experimental method. For example, line 205, culturing in what agar? Line 195, what is the size for hilA gene PCR product (bp)? Does it need further sequencing after PCR? Line 228, what is these standard or threshold to determine susceptible (S), susceptible with increased exposure (I), or resistant (R).

3. Please provide or specify more case using VITEK system in Introduction & Discussion part.

Author Response

Comments 1: Please specify the Table 1 data, antimicrobial susceptibility profile of 55 Salmonella strains should have a standard error if these MIC values are presented as average number.

Response 1: We thank the reviewer for this important observation. We would like to clarify that the data in Table 1 are presented as the Minimum Inhibitory Concentration (MIC) values obtained directly from the VITEK®2 Compact system, along with their categorical interpretation (S, I, R) according to BrCAST guidelines. The MIC is defined as the lowest concentration of an antimicrobial agent that prevents visible growth of the microorganism and therefore represents a single observed value for each isolate, not a mean value derived from multiple replicates. For this reason, it is not statistically appropriate to calculate a standard error or standard deviation for MIC results.

To avoid any misunderstanding, we have expanded the Results section (lines 145–159) to state explicitly that the values in Table 1 represent single MIC observations with categorical interpretations, not averaged data. This addition ensures clarity for readers and aligns the presentation of results with the methodology employed.

Comments 2: Section 4, please provide more detailed information for experimental method. For example, line 205, culturing in what agar? Line 195, what is the size for hilA gene PCR product (bp)? Does it need further sequencing after PCR? Line 228, what is these standard or threshold to determine susceptible (S), susceptible with increased exposure (I), or resistant (R).

Response 2: We thank the reviewer for this valuable observation. In the revised manuscript, we expanded the methodological description in Sections 4.1 and 4.3. Specifically, we: i) Added the selective enrichment media used (Buffered Peptone Water, Rappaport-Vassiliadis, and Tetrathionate broths), as well as the selective agars (XLD and BGA) and biochemical confirmation tests (Triple Sugar Iron Agar, TSI, and Lysine Iron Agar, LIA) (lines 296–306); ii) Clarified that isolates were confirmed at the genus level by PCR targeting the hilA gene (497 bp) (lines 304–305), and that no sequencing was performed since the assay was used exclusively for confirmation; iii) Specified the procedure for reactivation and maintenance of the strains (Brain Heart Infusion broth and Nutrient Agar) (lines 327–328); and iv) Expanded Section 4.3 to state that antimicrobial susceptibility testing was interpreted according to the Brazilian Committee on Antimicrobial Susceptibility Testing (BrCAST, 2023), harmonized with EUCAST, and included the categorical thresholds (S, I, R) as well as the specific breakpoints for each antimicrobial agent tested (lines 352–361).

We also emphasize that the full methodological details of the isolation, confirmation, and sampling procedures are available in the doctoral thesis cited as reference [62] (lines 295–296, and 323–324), which is openly accessible through the link provided in the article’s Data Availability Statement (lines 415–418).

Comments 3: Please provide or specify more case using VITEK system in Introduction & Discussion part.

Response 3: We thank the reviewer for this helpful suggestion. In the revised manuscript, we added context for the Vitek®2 Compact system in the Introduction (lines 113–117), citing both a foundational reference (Pincus, 2006) and a recent applied study involving foodborne Salmonella (Lozano-León et al., 2022), to highlight the relevance of this methodology. Furthermore, we strengthened the Discussion (lines 210–230) by incorporating additional studies that employed the Vitek®2 system, which allowed us to better contextualize and compare our findings.

Reviewer 2 Report

Comments and Suggestions for Authors

This study detected and analyzed the drug sensitivity of Salmonella in hybrid Tambaki fish raised in the state of Mato Grosso, Brazil. The topic of the paper has practical significance, which is related to the safety of aquatic food and public health, and corresponds to the issue of drug resistance that the World Health Organization is concerned about. The research design is clear and the methods are appropriate. The main finding is that no highly pathogenic serotype was detected and all isolates were sensitive to antibiotics. This "negative" result also has important reference value in the current context of rampant drug resistance.But there are several modification suggestions needed

1.Explain the possible reasons for the 'negative' result: The current discussion is somewhat weak. Suggest the author to delve deeper into why all strains are sensitive? This may be related to the usage regulations of antibiotics in fish farming in the region of Brazil? Or is it related to the specific sources and adaptability of Salmonella in these aquatic environments? Even without a definite answer, proposing these hypotheses can enrich the depth of the discussion.

2. Compared with similar studies, there is a need for a broader discussion of other studies on Salmonella resistance in aquatic environments or aquatic animals. Have other studies reported similar sensitive results? Or are there more reports of drug resistance? Is the result of this study a special case or a general phenomenon? By comparison, the significance of the findings in this study can be better identified.

3. Clarify the serotype detection method: The method section mentions that it aims to detect "clinically significant serotypes", but VITEK ®  The 2-system is typically used for species identification and drug susceptibility testing, rather than precise serum typing. Please clarify how serotypes such as S. Typhi and S. Paratyphi are excluded? Is it based on the biochemical identification results of the system, or was specific serological or molecular methods (such as PCR) used later? This point must be clarified, otherwise it will become the main point of doubt for the reviewers.

4. 

Refining Method Section:

Sample information: It is recommended to supplement the original sample information of 55 strains of bacteria (such as how many individuals did the fish come from? Specific tissue locations? Sampling time span?), which will help readers evaluate the representativeness of the results.

Drug susceptibility standards: Clearly state which standard (such as CLSI or EUCAST) is used to interpret drug susceptibility results

Author Response

Comments 1: Explain the possible reasons for the 'negative' result: The current discussion is somewhat weak. Suggest the author to delve deeper into why all strains are sensitive? This may be related to the usage regulations of antibiotics in fish farming in the region of Brazil? Or is it related to the specific sources and adaptability of Salmonella in these aquatic environments? Even without a definite answer, proposing these hypotheses can enrich the depth of the discussion.

Response 1: We thank the reviewer for this insightful comment. In the revised manuscript, we expanded the Discussion (lines 195–269) to include hypotheses that may explain the absence of antimicrobial resistance among the isolates. In particular, we added a paragraph addressing the ecological characteristics of freshwater environments and we cited the work of Deaven et al. (University of Georgia), which reported similar findings (lines 195–200). These additions help to contextualize our results and provide plausible explanations for the observed susceptibility.

Comments 2: Compared with similar studies, there is a need for a broader discussion of other studies on Salmonella resistance in aquatic environments or aquatic animals. Have other studies reported similar sensitive results? Or are there more reports of drug resistance? Is the result of this study a special case or a general phenomenon? By comparison, the significance of the findings in this study can be better identified.

Response 2: We appreciate the reviewer’s valuable suggestion. In the revised manuscript, we expanded the Discussion (lines 195–269) by incorporating studies on aquaculture and co-production systems, as well as by referencing our own previous work, which reported susceptibility yet was limited to a single slaughterhouse. By contrasting that earlier finding with the present study-covering multiple farms and a larger number of isolates, we provide a broader and more representative perspective, reinforcing the significance of our results.

Comments 3: Clarify the serotype detection method: The method section mentions that it aims to detect "clinically significant serotypes", but VITEK ®  The 2-system is typically used for species identification and drug susceptibility testing, rather than precise serum typing. Please clarify how serotypes such as S. Typhi and S. Paratyphi are excluded? Is it based on the biochemical identification results of the system, or was specific serological or molecular methods (such as PCR) used later? This point must be clarified, otherwise it will become the main point of doubt for the reviewers.

Response 3: We are grateful to the reviewer for highlighting this methodological point. We acknowledge that the VITEK®2 Compact system provides biochemical identification that may presumptively indicate certain serovars (e.g., S. Enteritidis, S. Typhimurium, S. Paratyphi), but full serotype confirmation requires complementary serological testing, which was not performed in this study. Accordingly, all isolates remained classified at the genus level (Salmonella spp.).

To ensure clarity and consistency, we have revised the Methods (lines 296–362) and Results section (lines 125–159) to state that no isolates were presumptively identified by the VITEK®2 system as these clinically relevant serotypes, while emphasizing that confirmatory serotyping was not performed. We also highlight that the genus-level identification provided by VITEK®2 was consistent with the previous confirmation of these isolates by PCR targeting the hilA gene (lines 131–133), as described in the Methods section (lines 304–305). This clarification aligns the Results with the methodological description.

Comments 4: Refining Method Section:

Comments 4.1 - Sample information: It is recommended to supplement the original sample information of 55 strains of bacteria (such as how many individuals did the fish come from? Specific tissue locations? Sampling time span?), which will help readers evaluate the representativeness of the results.

Response 4.1: We thank the reviewer for this important observation. As suggested, we expanded Section 4.1 to include the number of fish sampled (72 tambatinga), the specific tissues analyzed (scales, gills, muscle, and viscera, including esophagus, stomach, liver, intestine, and feces), and the sampling period (2022–2023, covering both rainy and dry seasons) (lines 296–324).

We also added the origin of the isolates, noting that they were collected from 25 fish farms across eight municipalities in the Baixada Cuiabana region, spanning both the Cerrado and Pantanal biomes. These additions improve clarity and allow readers to better evaluate the representativeness of the results. We further note that the doctoral thesis cited as reference [62] provides the full methodological details (lines 296 and 324); a link to this work is openly available in the manuscript’s Data Availability Statement (lines 415–418).

Comments 4.2 - Drug susceptibility standards: Clearly state which standard (such as CLSI or EUCAST) is used to interpret drug susceptibility results.

Response 4.2: We thank the reviewer for this important observation. The antimicrobial susceptibility testing in this study was interpreted according to BrCAST (2023) guidelines, which are harmonized with EUCAST (European Committee on Antimicrobial Susceptibility Testing). These guidelines define thresholds for categorizing isolates as susceptible (S), susceptible with increased exposure (I), or resistant (R). BrCAST has been recognized as the National EUCAST Committee since 2016, and EUCAST-based methods became mandatory in Brazil in 2018, ensuring alignment with international standards.

In the revised manuscript, we explicitly list the breakpoints applied to each antibiotic in the Methods section (4.3, lines 350–362), thereby ensuring that the standards used are clearly reported and consistent with international practice.

Round 2

Reviewer 1 Report

Comments and Suggestions for Authors

The authors addressed all my suggestions, good job.

I think the revised manuscript can be acceptable.

Reviewer 2 Report

Comments and Suggestions for Authors

I think this paper can be accepted.